# IL-18 binding protein can be a prognostic biomarker for idiopathic pulmonary fibrosis

Yu Nakanishi[1], Yasushi Horimasu[1]*, Kakuhiro Yamaguchi[1], Shinjiro Sakamoto[1‡], Takeshi Masuda[1‡], Taku Nakashima[1‡], Shintaro Miyamoto[1], Hiroshi Iwamoto[1], Shinichiro Ohshimo[2], Kazunori Fujitaka[1‡], Hironobu Hamada[3], Noboru Hattori[1‡]

1 Department of Molecular and Internal Medicine, Graduate School of Biomedical and Health Sciences, Hiroshima University, Minami-ku, Hiroshima, Japan, 2 Department of Emergency and Critical Care Medicine, Graduate School of Biomedical and Health Sciences, Hiroshima University Hospital, Minami-ku, Hiroshima, Japan, 3 Department of Physical Analysis and Therapeutic Sciences, Graduate School of Biomedical and Health Sciences Hiroshima University Hospital, Minami-ku, Hiroshima, Japan

☯ These authors contributed equally to this work.
‡ These authors also contributed equally to this work.
* yasushi17@hiroshima-u.ac.jp

**Data Availability Statement:** All relevant data is within the manuscript and its Supporting Information files.

## Abstract

Idiopathic pulmonary fibrosis is a chronic, fibrosing interstitial pneumonia that presents with various clinical courses and progression ranging from rapid to slow. To identify novel bio-markers that can support the diagnosis and/or prognostic prediction of idiopathic pulmonary fibrosis, we performed gene expression analysis, and the mRNA of interleukin-18 binding protein was increasingly expressed in patients with idiopathic pulmonary fibrosis compared with healthy controls. Therefore, we hypothesized that the interleukin-18 binding protein can serve as a diagnostic and/or prognostic biomarker for idiopathic pulmonary fibrosis. We investigated the expression of interleukin-18 binding protein in lung tissue, bronchoalveolar lavage fluid, and serum. Additionally, the correlation between interleukin-18 binding protein expression levels and the extent of fibrosis was investigated using mouse models of lung fibrosis induced by subcutaneous bleomycin injections. Serum interleukin-18 binding protein levels were significantly higher in idiopathic pulmonary fibrosis patients (5.06 ng/mL, inter-quartile range [IQR]: 4.20–6.35) than in healthy volunteers (3.31 ng/mL, IQR: 2.84–3.99) (p < 0.001). Multivariate logistic regression models revealed that the correlation between serum interleukin-18 binding protein levels and idiopathic pulmonary fibrosis was statistically independent after adjustment for age, sex, and smoking status. Multivariate Cox proportional hazard models revealed that serum interleukin-18 binding protein levels were predictive of idiopathic pulmonary fibrosis disease prognosis independent of other covariate factors (hazard ratio: 1.655, 95% confidence interval: 1.224–2.237, p = 0.001). We also demonstrated a significant positive correlation between lung hydroxyproline expression levels and interleukin-18 binding protein levels in bronchoalveolar lavage fluid from bleomycin-treated mice (Spearman r = 0.509, p = 0.004). These results indicate the utility of interleukin-18 binding protein as a novel prognostic biomarker for idiopathic pulmonary fibrosis.

**Funding:** YH has received research funding from Japan Society for the Promotion of Science (https://www.jsps.go.jp/, KAKENHI Grant No. 19K17676). The funders had no role in study design, data collection and analysis, decision to publish, or preparation of the manuscript.

**Competing interests:** The authors have declared that no competing interests exist.

## Introduction

Idiopathic pulmonary fibrosis (IPF) is a chronic, progressive, and fibrosing interstitial pneumonia of unknown cause, characterized by persistently progressive fibrosis of the lung interstitium that results in irreversible destruction of the alveolar structure [1]. The median survival of IPF patients has been reported to be approximately 3 years [1]. The clinical course of IPF varies widely from extremely rapid progression to relatively stable [2].

Pirfenidone and nintedanib have been approved as effective antifibrotic agents for IPF, although their efficacy is limited to modification of the extent of pulmonary functional deterioration [3–7]. With no available treatment option that can restore the deteriorated pulmonary function, early diagnosis of IPF as well as accurate prediction of disease progression are quite important for the clinical management of IPF [8, 9].

Biomarkers are commonly defined as objectively measurable indicators of the physiological and/or pathological processes of the organs or the response to therapeutic interventions [10]. For the reasons described above, substantial efforts have been made to identify a biomarker for IPF that can support the diagnosis, prognosis prediction, and the assessment of response to treatment [11]. We have previously performed gene expression analysis and have identified several molecules as potential biomarkers for IPF [12]. Among these, we focused on interleukin-18 binding protein (IL-18BP), which is known to be a decoy receptor for IL-18 [13]. In past reports, IL-18 was upregulated in patients with IPF and the lungs of bleomycin (BLM)-injured mice [14–16], but the role of IL-18BP, which is a natural antagonist of IL-18, is still unknown. Recently, it was reported that administration of IL-18BP to a BLM-injury model improved lung fibrosis [14, 17], but the trend of IL-18BP secreted *in vivo* is unclear. Therefore, the present study was conducted to clarify whether IL-18BP can serve as a diagnostic and/or prognostic biomarker for IPF.

## Methods

### Immunohistochemical staining for IL-18BP

Lung tissue sections were obtained from nine patients with IPF who agreed to undergo diagnostic surgical lung biopsies and provided written informed consent and permission to use their samples. The clinical characteristics of these patients were presented in S1 Table. Control lung tissue sections were obtained from healthy areas in the lungs of five patients with lung cancer who had undergone therapeutic surgical resection. The lung tissue sections were stained with ENVISION+ Kit/horseradish peroxidase (HRP) (Dako, Tokyo, Japan) as previously described [12]. Rabbit anti-IL-18BP antibody (ab52914, 1:3000, Lot ID: 161918; Abcam, Cambridge, UK) or rabbit control IgG antibody (ab37415, Lot ID: 3237588–1; Abcam) was added after blocking endogenous peroxidase and proteins. The sections were then incubated with HRP-labelled anti-rabbit IgG antibody followed by the addition of substrate-chromogen and counterstaining with hematoxylin.

In the nine IPF patients and five controls, the positively stained area was quantified using the ImageJ Fiji (National Institutes of Health, USA) software. We randomly selected five fields for each slide, and the average of the percentage of positively stained area in each of the five fields was deemed to be representative of the positively stained area for a patient.

### Subjects for Enzyme-Linked Immunosorbent Assay (ELISA) measurements

Between November 2001 and February 2017, 86 consecutive Japanese patients newly diagnosed with stable IPF at Hiroshima University Hospital were recruited, and serum and bronchoalveolar lavage fluid (BALF) samples were obtained. Patients who developed acute

exacerbation of IPF within 1 month of diagnosis or had lung cancer at the time of IPF diagnosis were excluded. IPF was diagnosed in accordance with the international diagnostic criteria published in 2018 [18]. Two hundred and fifty healthy subjects who underwent health check-ups were also enrolled as healthy volunteers (HVs) and serum samples were obtained. Each HV underwent pulmonary function tests and chest radiography studies, and those with apparent lung disease, such as interstitial lung disease or chronic obstructive pulmonary disease, were excluded. This study was approved by the Ethics Committees of Hiroshima University Hospital (IRB33) and conducted in accordance with the ethical standards established in the Helsinki Declaration of 1975. All patients and HVs provided informed consent in writing and permission to use their samples.

## Serum and BALF measurements

Serum samples, obtained from peripheral blood drawn during the initial assessment of patients with IPF or during the health check-ups for HVs, were stored at -80˚C until the time of analysis [19, 20]. Bronchoalveolar lavage was performed as previously described for patients with IPF [21]. Briefly, 50 mL of saline was introduced into the lung, promptly drawn, and this procedure was repeated three times [21]. The BALF was centrifuged promptly, and the supernatant was cryopreserved at -80˚C until analysis. BALF was available for 50 IPF patients. IL-18BP and IL-18 concentrations in the serum or BALF were measured using commercially available ELISA kits according to the manufacturer's instructions (R&D Systems, Minneapolis, MN, USA).

## Development of mouse models of lung fibrosis

Wild-type male C57BL/6 mice (Charles River Japan Inc., Yokohama, Japan) were housed in a specific pathogen free environment at an optimal temperature with a 12-h light/dark cycle and were used for experiments at 8–10 weeks of age. On day 0, osmotic minipumps (ALZET 1007D; DURECT, Cupertino, CA) containing either 100 μL of saline vehicle or bleomycin (100 mg/kg) and designed to deliver their contents at 0.5 μL/h for 7 days were implanted under the loose skin on the back of the mice slightly posterior to the scapulae under intraperitoneal anesthesia [22]. Combination anesthesia was prepared with 0.3 mg/kg of medetomidine, 4.0 mg/kg of midazolam, and 5.0 mg/kg of butorphanol. We monitored the mice in cages until they fully recovered from anesthesia without additional analgesics after osmotic pump implantation because the pump was small enough not to cause any obvious pain to the mice. The osmotic minipumps were removed on day 10 as recommended by the manufacturer. We monitored the body weight of the mice to evaluate health and welfare. Each experiment was performed using 5 or 6 mice based on the estimated mean and standard deviation of BALF IL-18BP in the preliminary experiment. One mouse in the bleomycin group died on day 21. At each time point, mice were euthanized by exsanguination. All mice were handled in accordance with Guidelines for Care and Use of Experimental Animals published by Hiroshima University, and all protocols were approved by the Hiroshima University (Approval #A19-34).

## Measurements of lung hydroxyproline, BALF IL-18BP and BALF IL-18

After tracheostomy, a 19G tube was inserted into the trachea, and the lungs were washed three times with 150 μL of saline vehicle. The recovered fluid was evaluated with a hemocytometer. The remaining BALF was centrifuged, and the supernatants were then collected and stored at -20˚C until ELISA was performed. The IL-18BP and IL-18 levels in BALF were measured using custom ELISA kits according to the manufacturer's instructions (Mouse IL-18 BPc ELISA Kit, RayBiotech, Peachtree Corners, GA, USA and Mouse IL-18 ELISA Kit, MBL,

Nagoya, Japan). The ELISA measurements were performed within 2 weeks of sample collection. At the time of sacrifice, the left lungs were removed and homogenized in phosphate buffered saline, followed by acid hydrolysis [23]. The collagen content was estimated by a colorimetric assay for hydroxyproline as described previously [23–25].

### Statistical analysis

Continuous variables are expressed as median and interquartile range (IQR). The differences between the two groups were tested using the Mann–Whitney U test or Pearson's chi-squared test, as appropriate, and those between three or more groups were tested using multiple Mann–Whitney U-tests with Bonferroni correction. Receiver operating characteristic (ROC) analysis was performed to assess the discriminating abilities. The area under the ROC curve (AUC) with 95% confidence interval (CI) was calculated to assess the discrimination power of serum IL-18BP and IL-18 levels. The correlations between two numerical variables were tested using Spearman correlations. Cox proportional hazards analysis was used to identify significant predictors of 3-year survival in IPF patients. Among the patients, those who had been treated with pirfenidone and/or nintedanib for 1 year or longer were classified as having received antifibrotic treatment. Survival was evaluated using the Kaplan–Meier approach, and the prognostic differences between the two groups were tested with the log-rank test. Those who died from non-IPF causes, including cancer, were treated as censored. Statistical analyses were performed using EZR (Saitama Medical Centre, Jichi Medical University, Saitama, Japan) for ROC analysis and JMP ®14 (SAS Institute Inc., Cary, NC, USA) for the other tests. A p value of $\leq 0.05$ was considered to be statistically significant for all analyses.

## Results

### Increased IL-18BP expression was detected in the lung tissue of IPF patients

In order to confirm our previous findings showing increased IL-18BP mRNA expression in IPF lungs [12], immunohistochemical staining for IL-18BP was performed. As shown in Fig 1A–1D, the IL-18BP expression in the lungs of representative three out of nine IPF patients was greater than that in the control lungs. The IPF lungs showed marked expression of IL-18BP in the fibrotic interstitium and bronchiole epithelial cells, as well as in the alveolar macrophages. Furthermore, as shown in S1 Fig, the percentage of the positively stained area was higher in patients with IPF (21.7% [IQR: 20.6–23.1]) than those in control group (10.6% [IQR: 7.5–13.0]) (p = 0.003).

### Increased serum IL-18BP levels were detected in IPF patients

To further assesses the clinical usefulness of IL-18BP as a biomarker, we also investigated serum IL-18 levels, which had previously been reported to be useful as a clinical biomarker for IPF [16], in both IPF patients and HVs. The characteristics of IPF patients and HVs are shown in Table 1. IPF patients were older and had history of heavier smoking than the HVs. As shown in Fig 2A, both serum IL-18BP and IL-18 levels were significantly higher in IPF patients than in HVs (p < 0.001 and p < 0.001, respectively). Importantly, ROC curve analysis revealed that the AUC value of serum IL-18BP was significantly larger than that of serum IL-18 for discriminating IPF patients from HVs (Fig 2B, p < 0.001).

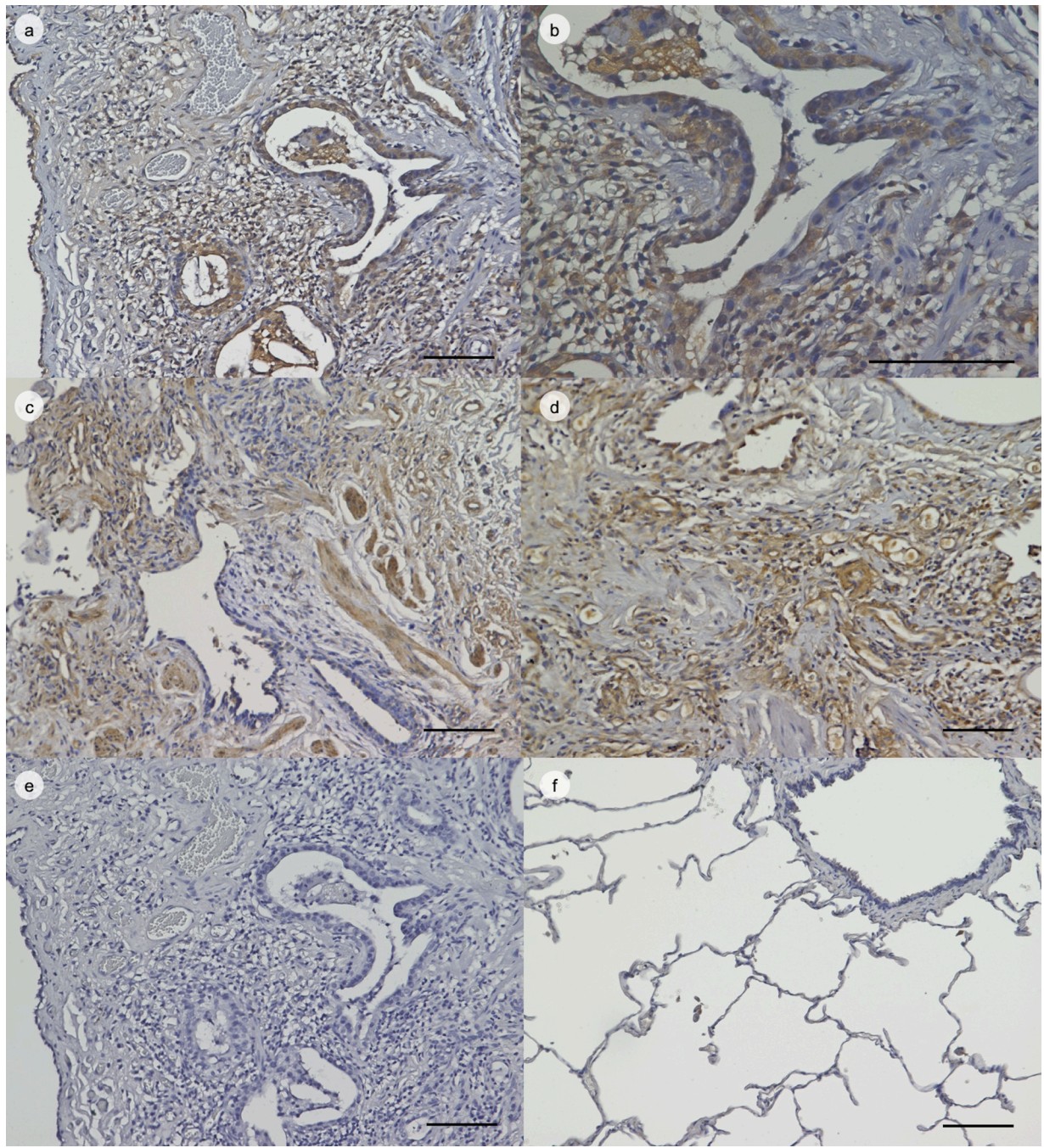

**Fig 1. Immunohistochemical expression of IL-18BP in IPF and control lungs.** (a, c, d) Immunohistochemical staining of IPF lung tissue with anti-IL-18BP antibody presented at x100 magnification. (b) The same field of view as in panel (a) presented at x400 magnification. (e) Immunohistochemical staining of IPF lung tissue with rabbit IgG presented at x100 magnification. (f) Immunohistochemical staining of control lungs with anti-IL-18BP antibody presented at x100 magnification. Scale bar = 100 μm. IL-18BP: interleukin-18 binding protein, IPF: idiopathic pulmonary fibrosis.

## Serum IL-18BP can independently predict the presence of IPF

Next, we investigated the correlations between serum IL-18BP and serum IL-18 or BALF IL-18BP levels. The correlations between serum IL-18BP and serum IL-18 levels were statistically

**Table 1. Patient characteristics.**

| Variables | IPF | HV | p-value |
|---|---|---|---|
| | N = 86 | N = 250 | |
| Age (years) | 68.5 (63.0–74.3) | 51.0 (43.0–55.3) | < 0.001 |
| Sex | | | |
| Male / Female | 71 / 15 | 213 / 37 | 0.604 |
| Smoking history | | | |
| Yes / No | 70 / 16 | 147 / 103 | < 0.001 |
| Pack-years | 30.8 (16.9–45.8) | 12.0 (0.0–32.0) | < 0.001 |
| Pulmonary function test | | | |
| FVC (% predicted) | 70.1 (61.5–81.8) | 98.4 (90.3–108.9) | < 0.001 |

Data are expressed as frequencies or medians (interquartile range).

IPF: idiopathic pulmonary fibrosis, HV: healthy volunteer, FVC: forced vital capacity.

significant not only in IPF patients (S2 Fig; r = 0.251, p = 0.019) but also in HVs (S2 Fig; r = 0.267, p < 0.001). In IPF patients, there were significant positive correlations between serum IL-18BP and BALF IL-18BP levels (S3 Fig; r = 0.406, p = 0.005).

In order to assess the statistical independence of the associations between the presence of IPF and serum IL-18BP or IL-18 levels, logistic regression analyses were performed. In the multivariate model, a high serum IL-18BP level was revealed to be a statistically independent predictor for the presence of IPF (p = 0.019, Table 2).

## High serum IL-18BP was associated with poor prognosis and impaired pulmonary function

As shown in Table 3, in IPF patients, the univariate Cox proportional hazards analysis revealed that increased serum IL-18BP levels, as well as deteriorated lung function was significantly associated with a poor prognosis. Additionally, the multivariate analysis confirmed that the serum IL-18BP level was a statistically independent predictor for the prognosis of IPF patients even when adjusted for age, sex, smoking history, percentage of forced vital capacity (%FVC), percentage of diffused capacity of the lung for carbon monoxide (%DLco), and the use of anti-fibrotic agents. Based on the ROC analysis shown in S4 Fig, we determined the optimal cut-off serum IL-18BP level that can discriminate those who died from those who did not, during 3 years from diagnosis, as 5.72 ng/mL. As shown in Fig 3A, the log-rank test revealed that participants with high serum IL-18BP levels had significantly poorer prognosis than those with low levels (p = 0.024). Furthermore, as shown in Fig 3B and 3C, serum IL-18BP levels showed a significant inverse correlation with %DLco (r = -0.300, p = 0.016), although it was not significantly correlated with %FVC (r = -0.177, p = 0.156).

## BALF IL-18BP levels were associated with the extent of lung fibrosis in the mice

To further investigate the associations between the expression levels of IL-18BP and the extent of lung fibrosis, we established a mouse model of lung fibrosis induced by subcutaneous bleomycin injections and investigated the associations between BALF IL-18BP levels and hydroxyproline expression in the lung (S5 Fig). As shown in Fig 4A, hydroxyproline expression in the lung tissue increased over time in the bleomycin- injected mice. On the other hand, BALF IL-18BP levels showed bimodal elevation with two peaks on day 7 and days 21 to 28 (Fig 4B). In

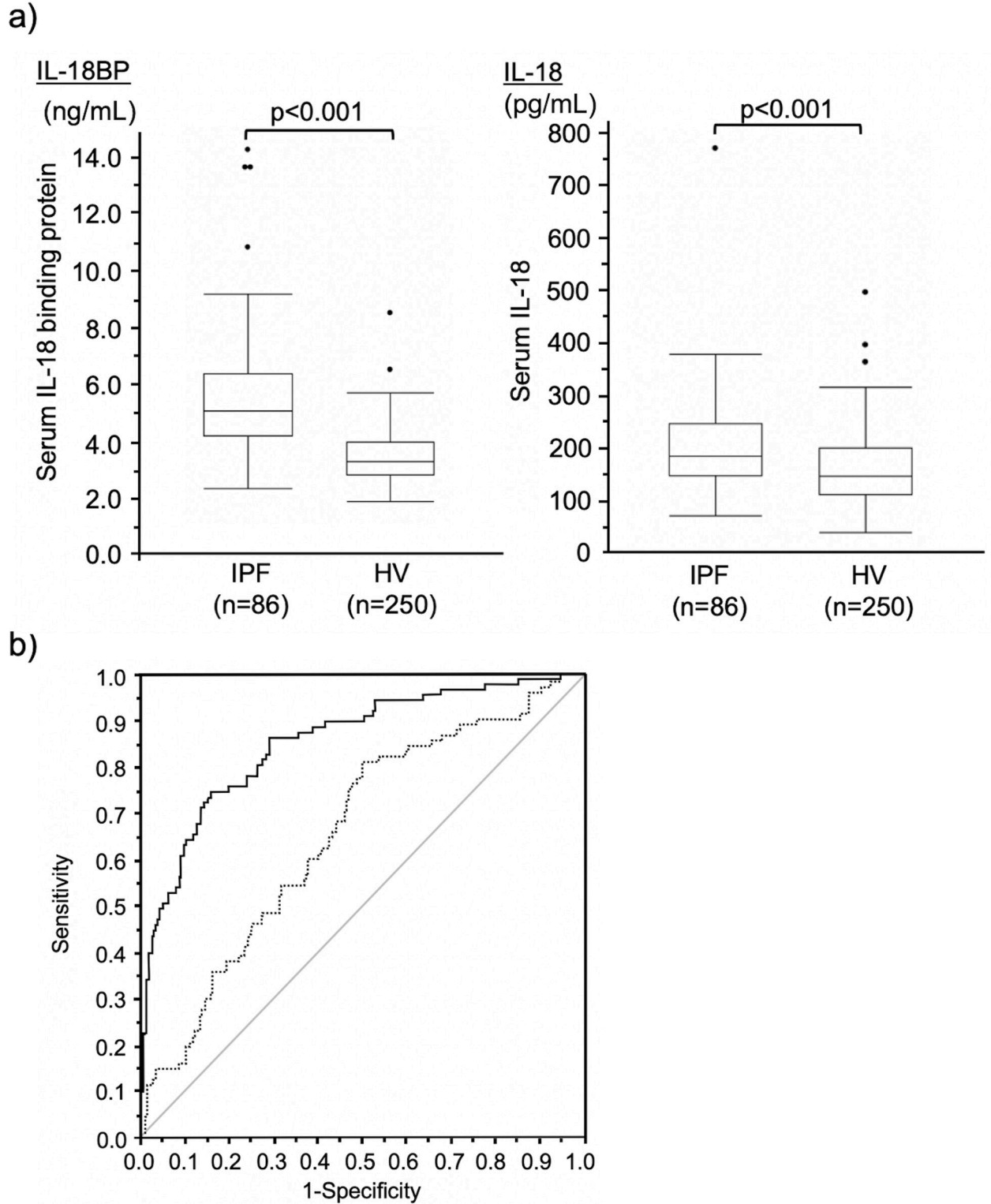

**Fig 2. Distribution of serum IL-18BP concentrations and ROC analysis in IPF patients and HVs.** (a) Box plot graphs showing ranges of serum IL-18BP and IL-18 levels in patients with IPF and HVs. The median IL-18BP levels of serum in patients with IPF and HVs were 5.06 (4.20–6.35) ng/mL and 3.31 (2.84–3.99) ng/mL (p < 0.001). The median IL-18 levels of serum in patients with IPF and HVs were 181.7 (146.6–243.6) pg/mL and 143.9 (108.5–196.8) pg/mL, respectively (p < 0.001). Boxes represent the 25th–75th percentiles; solid lines within the boxes show the median values, whiskers are the 10th and 90th percentiles, and the small dots represent outliers. (b) ROC curve analyses for serum IL-18BP (solid line) and IL-18

(dotted line) levels. The AUC value of serum IL-18BP (0.858, 95% Cl; 0.810–0.906) was significantly larger than that of serum IL-18 (0.661, 95%Cl: 0.595–0.726, p < 0.001). IL-18BP: interleukin-18 binding protein, ROC: receiver operating characteristics, IPF: idiopathic pulmonary fibrosis, HVs: healthy volunteers, IL-18: interleukin-18, AUC: area under the curve, CI: confidence interval.

addition, there were significant positive correlations between lung hydroxyproline and BALF IL-18BP levels in the bleomycin- injected mice (Fig 4C, r = 0.509, p = 0.004). Besides, BALF IL-18 levels showed unimodal elevation with a peak on day 7 (Fig 4D).

## Discussion

The aim of this study was to clarify whether IL-18BP can serve as a diagnostic and/or prognostic biomarker for IPF. As a result, we demonstrated that IL-18BP expression was increased in the lung tissue and serum of IPF patients and that the serum IL-18BP level was an independent prognostic predictor for IPF patients. We believe that these results are clinically important in that they show the usefulness of IL-18BP as a novel prognostic biomarker for IPF.

The most important finding in the present study was that the serum IL-18BP level could be a prognostic biomarker for IPF. In IPF patients, the expressions of IL-18BP both in the lung tissue and in the serum were increased (Figs 1 and 2), and increased serum IL-18BP levels were significantly associated with decreased %DLco (Fig 3C). Importantly, increased serum IL-18BP levels were significantly associated with poor prognosis of patients with IPF even in the multivariate model (Table 3). Furthermore, BALF IL-18BP levels in the mouse models of subcutaneously injected bleomycin-induced lung fibrosis were positively correlated with lung hydroxyproline expression levels (Fig 4). All of these findings suggest that increased expression of IL-18BP is associated with progressive lung fibrosis, pulmonary dysfunction, and poor prognosis in IPF patients. Additionally, these findings are consistent with the results of our previous gene expression study that demonstrated the increased expression of IL-18BP mRNA in the lung tissue of patients with IPF [12].

**Table 2. Logistic regression analysis of variables associated with the discriminating ability of IPF patients or healthy volunteers.**

|  | Odds ratio | 95% Cl | p-value |
|---|---|---|---|
| **Univariate analysis** |  |  |  |
| Age, years | 1.645 | 1.428–1.894 | < 0.001* |
| Sex, Male | 0.822 | 0.433–1.626 | 0.563 |
| Smoking history |  |  |  |
| Pack-years | 1.029 | 1.018–1.040 | 0.001* |
| FVC (%predicted) | 0.910 | 0.887–0.931 | 0.001* |
| Serum IL-18BP (ng/mL) |  |  |  |
| Continuous | 3.561 | 2.596–4.885 | < 0.001* |
| Serum IL-18 (pg/mL) |  |  |  |
| Continuous | 1.007 | 1.004–1.011 | 0.001* |
| **Multivariate analysis‡** |  |  |  |
| IL-18BP (ng/mL) |  |  |  |
| Continuous | 1.791 | 1.061–3.024 | 0.019* |

* p < 0.05 (logistic regression analysis).

‡Adjusted for age, sex, smoking history, and FVC (%predicted).

IPF: idiopathic pulmonary fibrosis, 95% Cl: 95% confidence interval, FVC: forced vital capacity, IL-18BP: interleukin-18 binding protein, IL-18: interleukin-18.

**Table 3. Prediction values for 3-year mortality in IPF patients assessed by Cox proportional hazards model (n = 86).**

|  | HR | 95% Cl | p-value |
|---|---|---|---|
| **Univariate analysis** |  |  |  |
| Age, years | 0.967 | 0.924–1.013 | 0.154 |
| Sex, male | 1.841 | 0.643–7.792 | 0.283 |
| Smoking history |  |  |  |
| Pack-years | 0.994 | 0.979–1.007 | 0.386 |
| FVC (%predicted) | 0.958 | 0.932–0.983 | < 0.001* |
| DLco (%predicted) | 0.926 | 0.888–0.964 | < 0.001* |
| Use of anti-fibrotic agent | 2.387 | 1.082–5.265 | 0.031* |
| Serum IL-18BP (ng/mL) |  |  |  |
| Continuous | 1.185 | 1.018–1.343 | 0.031* |
| BALF IL-18BP (pg/mL) |  |  |  |
| Continuous | 0.993 | 0.975–1.008 | 0.394 |
| Serum IL-18 (pg/mL) |  |  |  |
| Continuous | 1.000 | 0.996–1.003 | 0.897 |
| **Multivariate analysis** |  |  |  |
| Serum IL-18BP[‡] |  |  |  |
| Continuous | 1.655 | 1.224–2.237 | 0.001* |

* $p < 0.05$ (Cox proportional hazards model).

‡ Adjusted for age, sex, smoking history, FVC (%predicted), DLco (%predicted), and use of anti-fibrotic agent.

IPF: idiopathic pulmonary fibrosis, HR: Hazard ratio, 95% Cl: 95% confidence interval, FVC: forced vital capacity, DLco: diffusion capacity of the lung for carbon monoxide, IL-18BP: interleukin-18 binding protein, BALF: bronchoalveolar lavage fluid, IL-18: interleukin-18.

Although there are limited reports focusing on the utility of IL-18BP as a biomarker, Ha and colleagues reported that increased levels of serum IL-18BP were associated with increased mortality in total-body gamma irradiation mouse models [26]. We believe that this previous report supports our results. Although the mechanisms underlying the association between the increased expression of IL-18BP and the poor outcome of IPF remain unclear, we considered two possibilities. First, IL-18BP itself is not the cause of the poor outcomes for IPF; although IL-18BP itself has anti-fibrotic and/or anti-inflammatory activity, its increased expression simply reflects increased expression of IL-18, which can promote lung fibrosis. IL-18 is known to have high expression in the fibrotic lung [15, 16], and bleomycin-induced lung fibrosis can be altered by the inhibition of IL-18 expression [15]. Furthermore, IL-18 promotes the proliferation, migration, and collagen production of cardiac fibroblasts [27–31]. It is also known to promote the expression of IL-18BP by the production of interferon-γ [32–35]. IL-18BP is known to be a decoy receptor for IL-18 with substantially higher affinity than IL-18Rα, which is the functional receptor [13]. IL-18BP is known to act as a down-regulator for Th1 responses by reducing the induction of interferon-γ via inhibition of IL-18 activity [36]. Zhang et al. reported that exogenous administration of IL-18BP could attenuate both lipopolysaccharide-induced lung injury and bleomycin-induced lung fibrosis in murine models [14, 37]. In addition, IL-18BP was also reported to attenuate renal and hepatic injury in ischemia–reperfusion murine models through its antioxidant and anti-inflammatory activities [38]. Based on these reports, we can speculate that the increased expression of IL-18BP in IPF patients was not the cause of fibrosis, but a result of the increased expression of IL-18, a profibrotic factor. As shown in Fig 4, the unimodal elevation of BALF IL-18 followed by the bimodal elevation of

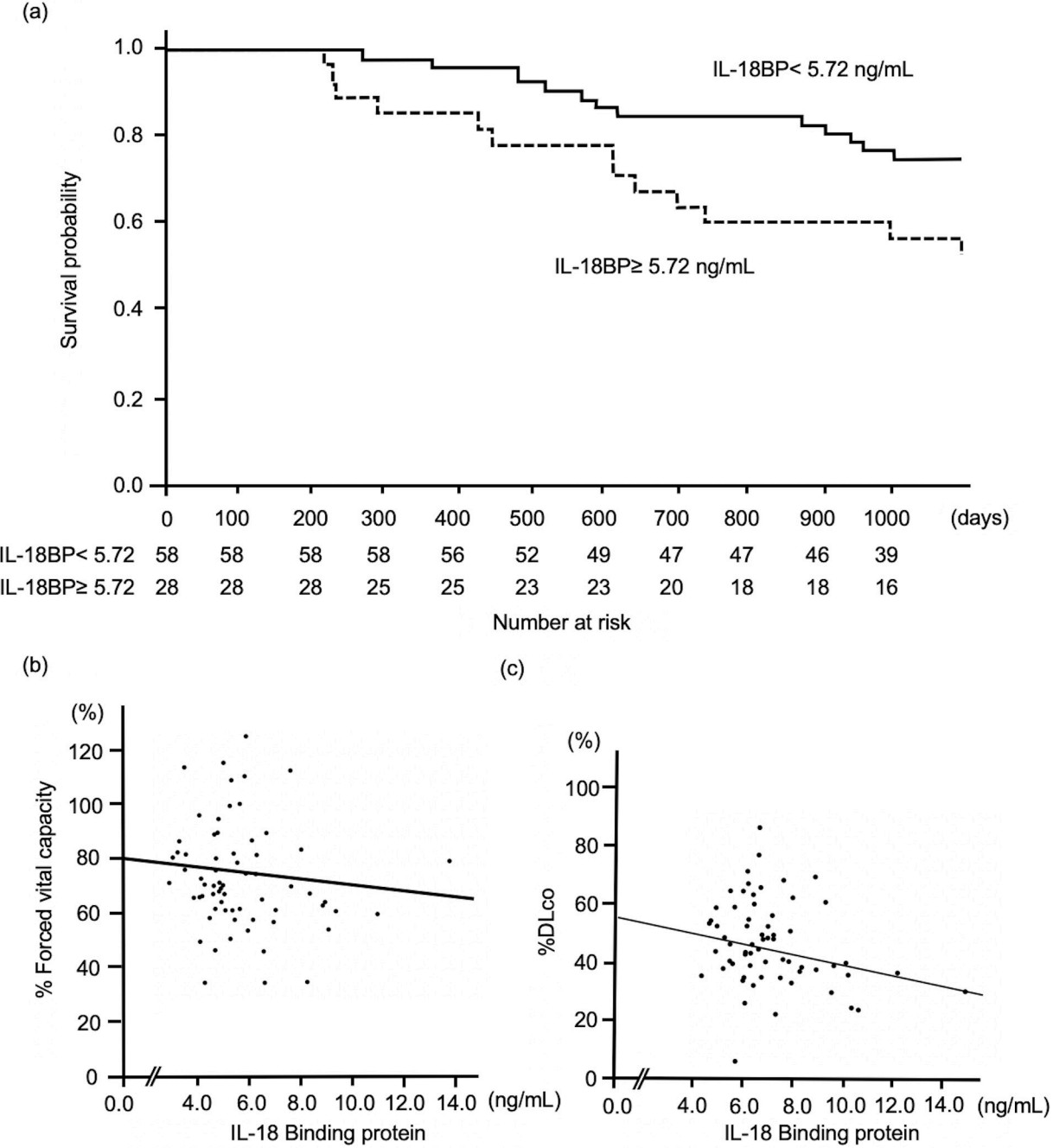

**Fig 3. Correlations between serum IL-18BP levels and pulmonary function or survival outcome of patients with IPF.** (a) Kaplan–Meier analysis for 3-year survival in patients with IPF, based on serum IL-18BP levels. Patients with higher levels of serum IL-18BP ($\geq$ 5.72 ng/mL) showed significantly poorer survival compared with those with lower serum IL-18BP (< 5.72 ng/mL). (b) Scatter plot graphs showing the correlations between serum IL-18BP levels and %FVC, which showed no significant correlation (Spearman r = -0.177, p = 0.156). (c) Serum IL-18BP levels showed significant negative correlations with %DLco (Spearman r = -0.300, p = 0.016). IL-18BP: interleukin-18 binding protein, IPF: idiopathic pulmonary fibrosis, FVC: forced vital capacity, DLco: diffusion capacity of the lung for carbon monoxide.

BALF IL-18BP in our murine models, might support this speculation in part. Further, such different serial changes in expression levels of IL-18BP and IL-18 may result in the different utilities of IL-18BP and IL-18 in predicting the outcome in IPF.

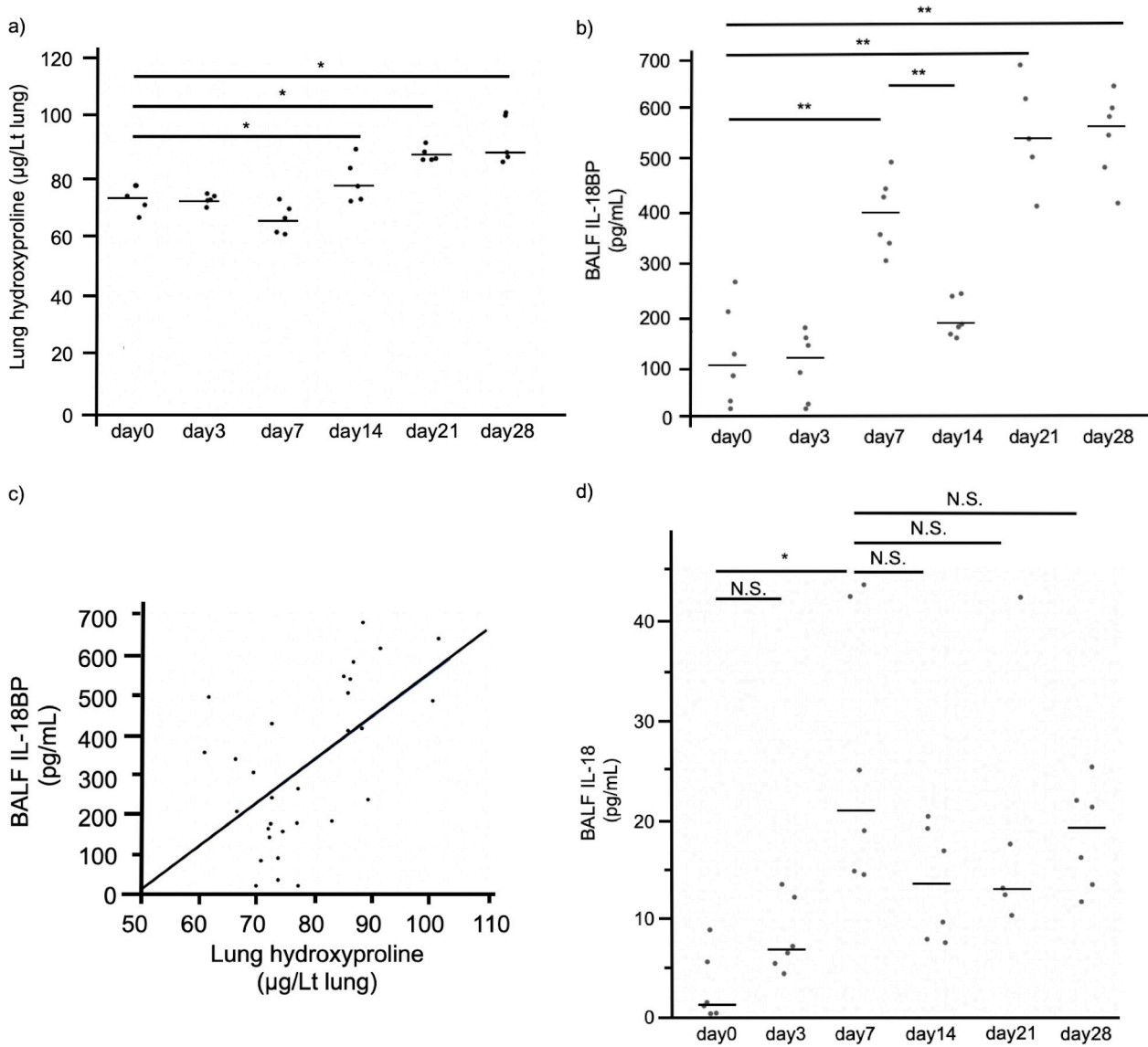

**Fig 4. Chronological change of BALF IL-18BP levels in the bleomycin-injected mice.** (a) Lung hydroxyproline levels chronologically increased over time in the bleomycin-injected mice. (b) BALF IL-18BP levels showed bimodal elevation with two peaks on day 7 and days 21 to 28 in the bleomycin-injected mice. The significance level was defined $\alpha = 0.013$ (four comparisons). (c) Lung hydroxyproline levels were significantly correlated with BALF IL-18BP levels (Spearman r = 0.509, p = 0.004). (d) BALF IL-18 levels showed unimodal elevation with a peak on day 7. The significance level was defined $\alpha = 0.010$ (five comparisons). Horizontal bars represent the median. BALF: bronchoalveolar lavage fluid, IL-18BP: interleukin-18 binding protein, IL-18: interleukin-18. * $p < 0.01$ (Mann–Whitney U-test with Bonferroni correction).

Another possibility is that IL-18BP promotes fibrosis via inhibition of interleukin-37 (IL-37), resulting in poor outcomes for IPF [39]. IL-37 has been reported to have a protective effect in interstitial lung disease through the inhibition of transforming growth factor-β1 signaling and the enhancement of fibroblast autophagy. IL-18BP has been reported to bind to IL-37, although the binding affinity is not high, and to suppress its activation, resulting in the promotion of fibrosis.

In the present study, there were significant positive correlations between serum IL-18BP and BALF IL-18BP levels in IPF patients. These results would indicate that the elevation of IL-18BP levels in sera of IPF patients reflects the elevation of IL-18BP levels in the alveolar space.

Under normal conditions, although relatively weak expression of IL-18BP mRNA was reported in the lung, protein expression has been reported in the appendix, spleen, tonsils, and lymph nodes, but not in the lung [40]. However, marked expression of IL-18BP mRNA has been reported in granulocytes, monocytes, and T-cells even under normal conditions [40]. Therefore, we can speculate that the migrated inflammatory cells can be the main source of IL-18BP expression in the IPF lung and that circulating IL-18BP levels might be also affected by its local expression in the lung. However, we didn't perform BAL in HVs from the view of medical ethics, so that we couldn't investigate the correlations between serum IL-18BP and BALF IL-18BP in HVs.

It might be surprising that the use of antifibrotic agents was significantly associated with a poor prognosis (Table 3). However, we consider that is caused by selection bias resulting from the physicians' and/or patients' decision, the medical insurance system, and the socio-economic status of the patient.

Although this study showed promising results, it had several limitations. First, the number of subjects included in the immunohistochemical analysis was relatively small. Second, there were significant differences in age and smoking history between HVs and patients with IPF. However, based on the results of the multivariate analyses (Tables 2 and 3), we believe that these differences did not diminish the correlation between IL-18BP and IPF. Third, this study was retrospective, and there was no validation cohort. Thus, multicenter prospective studies are required to confirm the usefulness of IL-18BP as a prognostic biomarker in IPF patients. Fourth, the influence of cryopreservation on the concentration of IL-18BP has not been directly assessed. The human samples have been preserved at -80˚C for several years, although the murine samples have been preserved at -20˚C for no longer than two weeks. Nevertheless, we believe that the measurement results were substantial because we found that they correlated with lung function and prognosis of patients with IPF. Furthermore, in the murine model, preliminary experiments revealed that serum samples stored at -20˚C and those stored at -80˚C showed almost comparable levels of IL-18BP. Therefore, we consider that cryopreservation does not significantly affect IL-18BP concentration. Finally, the interactions between IL-18BP and IL-18, or IL-37 were not directly investigated. Therefore, the detailed role of IL-18BP in the pathogenesis of IPF should be clarified in further investigations.

## Conclusion

This study showed that IL-18BP was overexpressed in the serum, BALF, and lung tissue of patients with IPF than in those of HVs, and that high serum IL-18BP concentrations were independently associated with poor prognosis in patients with IPF. This study is the first to demonstrate the usefulness of IL-18BP as a prognostic biomarker for IPF.

## Supporting information

**S1 Fig. Quantification of immunohistochemical staining with IL-18BP.** The percentage of the positively stained area was higher in patients with IPF (21.7% [IQR: 20.6–23.1]) than those in control lung (10.6% [IQR: 7.5–13.0]) (p = 0.003). Horizontal bars represent median. IL-18BP: interleukin-18 binding protein.
(TIF)

**S2 Fig. Correlations between serum IL-18BP levels and IL-18 levels.** (a) Serum IL-18BP and IL-18 levels showed significant positive correlations in HVs (Spearman r = 0.267, p < 0.001). (b) Serum IL-18BP and IL-18 levels showed significant positive correlations in IPF patients (r = 0.251, p = 0.019). IL-18BP: interleukin-18 binding protein, IL-18: interleukin-18, HVs:

healthy volunteers, IPF: idiopathic pulmonary fibrosis.
(TIF)

**S3 Fig. Correlation between serum IL-18BP levels and BALF.** Serum IL-18BP and BALF IL-18BP levels showed significant positive correlations in IPF patients (Spearman r = 0.406, p = 0.005). IL-18BP: interleukin-18 binding protein, BALF: bronchoalveolar lavage fluid, IPF: idiopathic pulmonary fibrosis.
(TIF)

**S4 Fig. ROC analysis of serum IL-18BP for 3-years survival in patients with IPF.** ROC analysis of serum IL-18BP levels was performed between those that have died and those that have survived or censored during three years from diagnosis (AUC 0.610, 95% Cl: 0.429–0.732). ROC: receiver operating characteristic, IL-18BP: interleukin-18 binding protein, IPF: idiopathic pulmonary fibrosis, AUC: area under the curve.
(TIF)

**S5 Fig. Summary of experimental model.** Eight-week-old male C57BL/6 mice were subcutaneously implanted with Alzet osmotic minipumps containing either a 200 μL saline vehicle or 100 mg/kg BLM at different doses. Pumps implanted under the back skin of mice slightly caudal to the scapulae were removed on day 10. BLM: bleomycin.
(TIF)

**S1 Table. Characteristics of IPF patients with lung tissue sample available.**
(DOCX)

## Acknowledgments

We are grateful to Dr. Nobuhisa Ishikawa in Hiroshima Prefectural Hospital and Dr. Masaya Taniwaki in Hiroshima Red Cross Hospital & Atomic-bomb Survivors Hospital who played the leading roles in gene expression analysis.

## Author Contributions

**Conceptualization:** Yasushi Horimasu, Hiroshi Iwamoto, Hironobu Hamada.

**Data curation:** Yu Nakanishi, Yasushi Horimasu, Kakuhiro Yamaguchi, Shinjiro Sakamoto, Takeshi Masuda, Taku Nakashima, Shintaro Miyamoto, Hiroshi Iwamoto, Shinichiro Ohshimo, Kazunori Fujitaka, Hironobu Hamada, Noboru Hattori.

**Formal analysis:** Yu Nakanishi, Yasushi Horimasu.

**Funding acquisition:** Yasushi Horimasu, Hiroshi Iwamoto, Noboru Hattori.

**Investigation:** Yu Nakanishi, Yasushi Horimasu, Kakuhiro Yamaguchi, Shintaro Miyamoto, Hiroshi Iwamoto, Shinichiro Ohshimo, Hironobu Hamada.

**Methodology:** Yu Nakanishi, Yasushi Horimasu, Kakuhiro Yamaguchi, Shintaro Miyamoto, Shinichiro Ohshimo, Hironobu Hamada.

**Project administration:** Yasushi Horimasu, Hiroshi Iwamoto, Hironobu Hamada, Noboru Hattori.

**Supervision:** Yasushi Horimasu, Hiroshi Iwamoto, Noboru Hattori.

**Writing – review & editing:** Yu Nakanishi, Yasushi Horimasu, Kakuhiro Yamaguchi, Shinjiro Sakamoto, Takeshi Masuda, Taku Nakashima, Shintaro Miyamoto, Hiroshi Iwamoto, Shinichiro Ohshimo, Kazunori Fujitaka, Hironobu Hamada, Noboru Hattori.

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
