## [Decision Letter · Decision Letter 0]

26 Feb 2021

PONE-D-20-35981

IL-18 binding protein can be a prognostic biomarker for idiopathic pulmonary fibrosis

PLOS ONE

Dear Dr. Horimasu,

Thank you for submitting your manuscript to PLOS ONE and please apologize the long review process. After careful examination of your manuscript by three experts in the field, we feel that it has merit but does not fully meet PLOS ONE’s publication criteria as it currently stands. Therefore, we invite you to submit a revised version of the manuscript that addresses the points raised during the review process.

All reviewers acknowledged that your manuscript is interesting and relevant, however, the reviewers ask you to be more precise and to clarify your statements. I also suggest to verify that all your conclusions are supported by your results. Otherwise they should be mentioned as hypothesis.

We look forward to receiving your revised manuscript.

Kind regards,

Gernot Zissel, Ph.D.

Academic Editor

PLOS ONE

2. Please address the following items with respect to animal research procedures and welfare: (1) clearly describe the post-operative analgesics and supportive care that you had provided to the animals following the osmotic pump implantation procedure. (2) Provide an explanation for how you arrived at 6 animals per group. Did you conduct a power analysis or any other type of statistical test; (3) state the rate of mortality during the experiment (if any); (4) discuss humane endpoints, that is: did you have a plan in place to euthanize animals who became severely ill during the study (prior to the experimental endpoints); (4) state the monitoring parameters for the animals (the clinical and behavioral signs used to evaluate the health and welfare). (5) State the method of euthanasia.

<h1>** **</h1>

Reviewers' comments:

Reviewer's Responses to Questions

**Comments to the Author**

1. Is the manuscript technically sound, and do the data support the conclusions?

Reviewer #1: Partly

Reviewer #2: Yes

Reviewer #3: Yes

2. Has the statistical analysis been performed appropriately and rigorously? 

Reviewer #1: Yes

Reviewer #2: Yes

Reviewer #3: I Don't Know

3. Have the authors made all data underlying the findings in their manuscript fully available?

Reviewer #1: Yes

Reviewer #2: Yes

Reviewer #3: Yes

4. Is the manuscript presented in an intelligible fashion and written in standard English?

Reviewer #1: Yes

Reviewer #2: No

Reviewer #3: Yes

5. Review Comments to the Author

Reviewer #1: In this manuscript Yu Nakanishi et al, entitled: “IL-18 binding protein can be a prognostic biomarker for idiopathic pulmonary fibrosis” investigated the role of IL-18 in the diagnosis and prognosis of IPF. The paper resulted of interest, however some point need to be improved:

major comments:

Introduction

-The phrase “The prognosis of IPF is known to be poorer than that of certain malignancies” resulted not appropriate in this context, I suggest to delete.

- “although their efficacy is limited to modification of the extent of pulmonary functional deterioration.” Miss references.

- “novel biomarkers that can support an early diagnosis of IPF and/or predict the

progression of the disease are urgently needed.” This sentence resulted vague. Please can the author specify the role of biomarker in IPF? What are the state of art about biomarker in IPF? I also suggest to better define the role of biomarker that the authors intend to emphasize, diagnostic? Prognostic? Response to treatment? Please improve this part of introduction.

- “We previously performed gene expression analysis using surgically resected lung

tissue from seven patients with IPF as well as five patients with non-specific interstitial

pneumonia [10].” Is redundant in the introduction, please move in the discussion section.

“Therefore, we performed the present study to clarify whether IL-18BPcan serve as a clinical biomarker for IPF.” Rephrase this sentence and better explain the purpose of the study

Materials and methods

- Authors excluded patients who develop lung cancer? Because 30% of patients with IPF develop lung within one year. Please clearify this point.

- “Bronchoalveolar lavage was performed as previously described.” Please add the reference

- “cryopreserved at -80℃ until analysis.” Can the concentration of IL-18BP change due to -80°C cryopreserved?

- “The remaining BALF was centrifuged, and the supernatants were then collected and stored at -20℃ until ELISA was performed.” Different method of storage were applied between BAL of Human and Mouse. Please can the authors clearify this discrepancy?

- Results are expressed as mean±SD, however the non parametric test revealed that the variables are not normally distributed. Please can the authors explain why they do not use median and IQR?

Results

- Age and smoking habits differred in the 2 population. What think the authors about this? Can these variable modify the results and the concentrations of IL18-IL18BP?

- “The correlations between serum IL-18BP and serum IL-18 levels were statistically significant not only in IPF patients but also in HVs (S1 Fig).” Please report r and p values.

- “ In IPF patients, there were significant positive correlations between serum IL-18BP and BALF IL-18BP levels (S2 Fig, r = 0.406, p = 0.005).” What think authors about this results?

- “Cox proportional hazards analysis revealed that increased serum IL-18BP levels, as well as low % forced vital capacity (FVC), % diffuse capacity of the lung for carbon monoxide (DLco), and use of antifibrotic agents was significantly associated with a poor prognosis.” You say that the use of antifibrotic agent promote the poor prognosis?

Discussion

- First of all clearify and better explain the aim

- “can serve as a clinical biomarker for IPF” what kind of biomarker? Dagnostic? Prognostic?

- “We believe that these results are clinically important in that they show the usefulness of IL-18BP as a novel predictive biomarker for IPF.” From clinical point of view this result does not help to discriminate IPF than other ILD were normally differential diagnosis occur, but differentiate from Controls, Rarely IPF patients were confused with Controls. Please reformulate.

- “We consider that this previous report supports our results, although the mechanisms underlying how IL-18BP causes a poor outcome in IPF remains unclear.” This is true but can the authors better explain what are the state of art about the knowledge of il-18 in the lung and the implications in ipf pathogenesis?

- “Despite these limitations, we believe that the results of our study are of significance in that we have demonstrated the utility of serum IL-18BP as a prognostic biomarker for IPF for the first time.” The final sentence resulted vague and inappropriate.

Conclusion

- The conclusions are inappropriate. Please better specify the role of our research avoiding vague sentences.

Reviewer #2: The manuscript “IL-18 binding protein can be a prognostic biomarker for idiopathic pulmonary fibrosis” by Nakanishi et al. explores the novel biomarker of IL-18BP as a prognostic indicator in IPF. It is proposed that this protein may play a role in the pathogenesis of the disease and can indicate the extent of fibrosis. In this paper IL-18BP is studied through analysis of human samples, including blood, BALF, and lung, and through murine bleomycin-induced models of pulmonary fibrosis. This is an important topic in the field as there is currently a lack of clinically relevant prognostic biomarkers in IPF, and the authors should be applauded for their work in addressing this matter. However, the manuscript would be improved with the consideration of the following comments:

Introduction

“Recently, it was reported that administration of IL- 18BP to a BLM-injury model improved lung fibrosis, but the trend of IL-18BP secreted in vivo is unclear.” Could the authors please provide a citation for the study that this statement applies to?

Methods

Immunohistochemical staining:

Could the authors please provide further information on the surgical lung biopsies? Were these tissues reviewed by a pathologist? What were the inclusion/exclusion criteria? How many tissues were collected? What dates were these tissues collected from? Are demographic/clinical characteristics available for these subjects?

Subjects for ELISA measurements:

Could the authors please clarify what samples were collected from these subjects?

Results

Figure 1:

How many tissues were stained? It would be beneficial to include images from other IPF and control subjects as well, to confirm findings across multiple samples.

Please include a quantitative measure of staining to strengthen the findings.

Figure 3:

In panel c the authors state there is a positive correlation, but the plot demonstrates a negative correlation. Please adjust.

Discussion

The findings of this paper convey the significance of IL-18BP in both the circulatory and lung tissue compartments in IPF. Although the authors state it is expressed in bronchiole epithelial cells and alveolar macrophages in the IHC results, is there any existing literature on the cell types that IL-18BP is typically expressed in, in both the blood and lung tissue? It would be beneficial for the authors to include this, as well as how this may relate to the results of this study.

Conclusion

The authors state that IL-18BP “might be a potential therapeutic target of IPF.” It would be useful for the authors to further explain this in the discussion section, given what is known about its mechanism in conjunction with the findings of this manuscript.

General

The manuscript should be diligently edited for grammar and syntax errors.

Reviewer #3: This paper may be of interest to those researching IPF; a disease which requires better predictive biomarkers of disease prognosis.

In the introduction the authors explain and highlight the need for early diagnosis of IPF and the requirement for predictive biomarkers of prognosis for clinical management. They explain how they arrived at their hypothesis (from a previous publication by this group) that IL-18BP may serve as a clinical biomarker for IPF. The methods are sufficiently detailed.

The authors suggest that IL-18BP is elevated in IPF lung tissue compared to non-cancerous lung tissue from lung cancer patients. They address the limitation of this analysis due to the number of subjects being relatively small, however n numbers are not provided. N numbers should be provided in the methods, results or figure legend and it should be made clear that the images shown in figure 1 are representative of a larger cohort.

The authors show that both IL-18 and IL-18BP levels are increased in IPF patient serum compared to healthy controls and that serum IL-18BP may be more effective than IL-18 at discriminating IPF patients from healthy controls. The authors go on to show that serum IL-18 and IL-18BP levels correlate with each other in both IPF and healthy patients. The authors refer to ‘S1 fig’ to demonstrate this finding, however these graphs need titles to explain which refers to healthy and which refers to IPF data.

The authors show that IPF serum IL-18BP and IPF BALF IL-18BP levels also correlate with each other, however due to the lack of healthy BALF the same comparison could not made for healthy patients and this should be considered as a limitation when making any conclusions from this. However, this does not particularly affect the overall findings of this manuscript.

The authors perform univariate analysis to demonstrate that IL-18BP (as well as other factors) are statistically correlated with a poor prognosis defined as survival of less than 3 years from diagnosis. Indeed, further multivariate analysis suggests IL-18BP may be an independent predictor of prognosis independent of other listed factors. I am not an expert in these types of statistical analyses and so cannot comment on their robustness.

The authors describe in fig 3 that IL-18BP is inversely correlated with %DLco and %FVC which further supports IL-18BP as a biomarker of prognosis in IPF. In figure 3 they also use a Kaplan-Meier plot to illustrate that IPF patients with higher serum levels of IL-18BP (>5.72 ng/ml) showed poorer survival compared to those with lower IL-18BP levels. The authors refer to ‘S3 Fig’ to explain how they determined 5.72ng/ml as the optimal cut-off for predicting the 3 year survival of patients, however I do not understand how this figure explains this and further explanation is required.

Finally, the authors use a mouse model of lung fibrosis to corroborate their findings in humans, illustrating that increases in IL-18 and IL-18BP are associated with increased hydroxyproline expression in lung tissue. I question why they felt that they needed to use a mouse model to confirm what they had already demonstrated in humans. The advantage of using a mouse model here would be to try and understand a pathogenic role for IL-18BP in lung fibrosis however this was not done. For example, in the discussion they speculate two potential roles for IL-18BP in fibrosis; a pathogenic role via inhibition of IL-37 and a non-pathogenic role/protective role simply as a consequence of increased IL-18 expression. Both of these are valid hypotheses that could quite easily have been investigated in the mouse model used. Did the authors look for increased expression of IL-37 in this model to support this?

The authors conclude that IL-18BP is elevated in IPF and may be a predictive biomarker of prognosis. They go on to suggest that IL-18BP may be a key molecule in the development and progression of IPF and might be a potential therapeutic target. I don’t see any evidence in this paper that supports the claim that IL-18BP has a pathogenic role in IPF or that by inhibiting it would impact the course of disease (again where the mouse model could have been used to explore this mechanism). We know from experience that simply because something is elevated in disease it does not mean it is pathogenic or make it a therapeutic target. The known actions of IL-18BP could equally support an attempt by the body to protect against disease progression in patients with rapidly progressing disease.

Overall, the paper is sound with these minor revisions:

1) Include n numbers for figure 1 and explain images shown are representative.

2) Add graph titles to ‘S1 fig’ to discriminate between IPF and healthy controls.

3) Include lack of healthy BALF as a limitation to this study when discussing correlations between IL-18BP in serum an BALF.

4) Further explanation of how they determined 5.72ng/ml IL-18BP as the optimal cut-off for predicting the 3 year survival of patients, as it is not clear from ‘S3 fig’.

5) It is a shame that the mouse model wasn’t used to investigate the role of IL-18BP in fibrosis. Did the authors consider looking at IL-37 in this model to support their hypothesis in the discussion?

6) If the authors have any evidence to support a pathogenic role of IL-18BP in IPF it should be included to support the claim that “IL-18BP may be a key molecule in the development and progression of IPF and might be a potential therapeutic target of IPF” rather than it is simply elevated in disease and a potential clinical biomarker which is otherwise what they have demonstrated. Alternatively, make it clear that this is a hypothesis of this group and further experiments are required. Literature suggests that IL-18BP is potentially protective in fibrosis and its inhibition could actually exacerbate disease.

6. PLOS authors have the option to publish the peer review history of their article (what does this mean?). If published, this will include your full peer review and any attached files.

Reviewer #1: No

Reviewer #2: No

Reviewer #3: No

---

## [Author Response · Author response to Decision Letter 0]

31 Mar 2021

March, 29, 2021

PLOS ONE 

COMMENTS FOR THE AUTHOR:

PONE-D-20-35981

IL-18 binding protein can be a prognostic biomarker for idiopathic pulmonary fibrosis

Joerg Heber

Editors-in-Chief

PLOS ONE

Dear Editor:

We wish to re-submit the manuscript titled “IL-18 binding protein can be a prognostic biomarker for idiopathic pulmonary fibrosis.” The manuscript ID is PONE-D-20-35981.

We thank you and the reviewers for your thoughtful suggestions and insights. The manuscript has benefited from these insightful suggestions. I look forward to working with you and the reviewers to move this manuscript closer to publication in PLOS ONE.

The responses to all comments have been prepared and attached word file 'Revised Manuscript with Track Changes' and 'Manuscript'.

Thank you for your consideration. I look forward to hearing from you.

Sincerely,

Yasushi Horimasu

Department of Respiratory Internal Medicine, Hiroshima University Hospital

1-2-3 Kasumi, Minami-ku, Hiroshima 734-8551, Japan.

Tel: 082-254-8551

E-mail: yasushi17@hiroshima-u.ac.jp

First, we revised our manuscript in accordance with the following Journal requirements:

2. Please address the following items with respect to animal research procedures and welfare: (1) clearly describe the post-operative analgesics and supportive care that you had provided to the animals following the osmotic pump implantation procedure. (2) Provide an explanation for how you arrived at 6 animals per group. Did you conduct a power analysis or any other type of statistical test; (3) state the rate of mortality during the experiment (if any); (4) discuss humane endpoints, that is: did you have a plan in place to euthanize animals who became severely ill during the study (prior to the experimental endpoints); (4) state the monitoring parameters for the animals (the clinical and behavioral signs used to evaluate the health and welfare). (5) State the method of euthanasia.

(1) We monitored the mice in cages until they reached full recovery from anesthesia without additional analgesics after osmotic pump implantation, because the pomp was small enough not to cause any obvious pain to the mice. This point was clearly stated in the revised manuscript (Line 135). 

(2) We decided the number of animals based on the estimated mean and standard deviation of BALF IL-18BP in the preliminary experiment. This point was added in the revised manuscript (Line 139). 

(3) One mouse in the bleomycin group died on day21. This information was also included in the revised manuscript (Line 141).

(4) We didn’t have a plan in place to euthanize animals who became severely ill because no mouse died or got severely ill in the preliminary experiment. Additionally, this study was conducted in compliance with Guidelines for Care and Use of Experimental Animals published by Hiroshima University. Therefore, the description was added (Line 142).

(4) We monitored the body weight of the mice to evaluate the health and welfare. Therefore, the description was added (Line 138).

(5) The method of euthanasia was exsanguination after anesthesia. This statement was included in the revised manuscript (Line 142).

Next, responses for each reviewer's comment are as follows.

 Reviewer #1: 

major comments:

Introduction

COMMENT #1.

-The phrase “The prognosis of IPF is known to be poorer than that of certain malignancies” resulted not appropriate in this context, I suggest to delete.

RESPONSE #1.

We deleted this phrase in accordance with the reviewer’s comment (Lines 55).

COMMENT #2.

- “although their efficacy is limited to modification of the extent of pulmonary functional deterioration.” Miss references.

RESPONSE #2.

Thank you very much for your suggestion. We changed the insertion point of the references from the middle of the sentence to the end (Lines 60).

COMMENT #3.

- “novel biomarkers that can support an early diagnosis of IPF and/or predict the progression of the disease are urgently needed.” This sentence resulted vague. Please can the author specify the role of biomarker in IPF? What are the state of art about biomarker in IPF? I also suggest to better define the role of biomarker that the authors intend to emphasize, diagnostic? Prognostic? Response to treatment? Please improve this part of introduction.

RESPONSE #3.

We admit that the reviewer’s comment is critically important. In order to clearly state the role of biomarker in IPF and also to define the role of biomarker that we intend to emphasize, we altered the final paragraph of the Introduction section in the revised manuscript (Line 63 to 67).

COMMENT #4.

- “We previously performed gene expression analysis using surgically resected lung tissue from seven patients with IPF as well as five patients with non-specific interstitial pneumonia [10].” Is redundant in the introduction, please move in the discussion section.

RESPONSE #4.

In accordance with the reviewer’s recommendation, we simplified the sentence (Line 67) and briefly restated in discussion section (Line 333).

COMMENT #5.

- “Therefore, we performed the present study to clarify whether IL-18BP can serve as a clinical biomarker for IPF.” Rephrase this sentence and better explain the purpose of the study.

RESPONSE #5.

We agree with the reviewer's comment. With COMMENT #3 also in mind, we altered the last sentence in the Introduction section (Line 74).

Materials and methods

COMMENT #6.

- Authors excluded patients who develop lung cancer? Because 30% of patients with IPF develop lung within one year. Please clearify this point.

RESPONSE #6.

Based on the reviewer’s comment, we clarified that those with lung cancer at the time of IPF diagnosis were excluded in this study in “Subjects for enzyme-linked immunosorbent assay (ELISA) measurements” subsection (Line 102). Further, in the survival analysis, those who died from non-IPF causes, including cancer, were treated as censored. This statement was added in “Statistical analysis” subsection (Line 175).

COMMENT #7.

- “Bronchoalveolar lavage was performed as previously described.” Please add the reference

RESPONSE #7.

We added the reference number #21 in accordance with the reviewer’s comment (Line 118).

COMMENT #8.

- “cryopreserved at -80℃ until analysis.” Can the concentration of IL-18BP change due to -80°C cryopreserved?

COMMENT #9.

- “The remaining BALF was centrifuged, and the supernatants were then collected and stored at -20℃ until ELISA was performed.” Different method of storage were applied between BAL of Human and Mouse. Please can the authors clearify this discrepancy?

RESPONSE #8 and #9.

We preserved the human samples at -80℃ because they need to be preserved for several years. There are some reports demonstrating the successful measurement of IL-18BP with cryopreserved samples [1-2]. Although the influence of cryopreservation to the concentration of IL-18BP has not been directly assessed, we believe that the measurement results were substantial because we found they correlated with lung function and prognosis of patients with IPF. On the other hand, the storage period of murine samples was shorter than two weeks. Therefore, we planned to preserve murine samples at -20℃. We added these descriptions in Method section (Line 155). Furthermore, in the murine model, serum samples stored at -20℃ and those stored at -80℃ showed almost comparable levels of IL-18BP. Therefore, we consider that the cryopreservation does not significantly affect IL-18BP concentration. These points were included as the limitation in the revised manuscript (Line 393 to 401). 

[1] Mediators Inflamm. 2014; 2014: 165742.

[2] Int J Cancer. 2011; 129: 424–432.

COMMENT #10.

- Results are expressed as mean±SD, however the non parametric test revealed that the variables are not normally distributed. Please can the authors explain why they do not use median and IQR?

RESPONSE #10.

We admit that the reviewer's concern is reasonable. We changed the presentation of the numerical variable to median and IQR.

 (Line 37, 162, 189, 219, S1 Fig, and Table1).

Results

COMMENT #11.

- Age and smoking habits differred in the 2 population. What think the authors about this? Can these variable modify the results and the concentrations of IL18-IL18BP?

RESPONSE #11.

As the reviewer pointed out, age and smoking habits were different between HVs and patients with IPF. This is the case because HVs were mostly consisted of young office workers. To avoid or minimize the influence of these differences to our result, we had included age and smoking history in the multivariate model both in the logistic regression analysis (Table 2) and in the Cox analysis (Table 3). Furthermore, we also included these points in the limitation of the revised manuscript (Line 387 to 392).

COMMENT #12.

- “The correlations between serum IL-18BP and serum IL-18 levels were statistically significant not only in IPF patients but also in HVs (S1 Fig).” Please report r and p values.

RESPONSE #12.

Thank you for your suggestion. We reported r and p values (Line 234).

COMMENT #13.

- “ In IPF patients, there were significant positive correlations between serum IL-18BP and BALF IL-18BP levels (S2 Fig, r = 0.406, p = 0.005).” What think authors about this results?

RESPONSE #13.

We admit that the reviewer’s comment is critically important. We consider that this result would indicate that the elevation of IL-18BP levels in sera of IPF patients reflects the elevation of IL-18BP levels in the alveolar space. In the normal condition, the expression of IL-18BP was reported to be ubiquitous in the lymphoid tissue, although in those with IPF, circulating IL-18BP levels were mainly affected by its local expression in the lung. These points were addressed in the new paragraph in Discussion section (Line 370 to 382).

COMMENT #14.

- “Cox proportional hazards analysis revealed that increased serum IL-18BP levels, as well as low % forced vital capacity (FVC), % diffuse capacity of the lung for carbon monoxide (DLco), and use of antifibrotic agents was significantly associated with a poor prognosis.” You say that the use of antifibrotic agent promote the poor prognosis?

RESPONSE #14.

We agree that this sentence is quite miss-leading. we consider that this “ostensible” association between use of antifibrotic agents and poor prognosis was caused by the selection bias resulting from physicians’ and/or patients' decision, medical insurance system and economical status of the patients. Therefore, we altered the description in Result section (Line 254), and inserted the interpretation of this “ostensible” association in Discussion section (Line 383 to 386).

Discussion

COMMENT #15.

- First of all clearify and better explain the aim

RESPONSE #15.

We admit that the beginning sentence of Discussion section was roundabout. In accordance with the reviewer’s comment, we modified the description of the aim of the study in the beginning of discussion (Line 317).

COMMENT #16.

- “can serve as a clinical biomarker for IPF” what kind of biomarker? Dagnostic? Prognostic?

RESPONSE #16.

Our initial hypothesis was that IL-18BP can serve as both diagnostic and prognostic biomarker. Therefore, the aim of this study was to clarify whether IL-18BP can serve as a diagnostic and/or prognostic biomarker for IPF. We have clearly describe this aim in the revised manuscript (Line 320).

COMMENT #17.

- “We believe that these results are clinically important in that they show the usefulness of IL-18BP as a novel predictive biomarker for IPF.” From clinical point of view this result does not help to discriminate IPF than other ILD were normally differential diagnosis occur, but differentiate from Controls, Rarely IPF patients were confused with Controls. Please reformulate.

RESPONSE #17.

I think the reviewer's concerns are quite important. In the present study, IL-18BP was shown to be useful in prognosis prediction of IPF, although its usefulness in differential diagnosis of ILDs was not demonstrated. Therefore, we changed the words “predictive biomarker” to “prognostic biomarker” (Line 320).

COMMENT #18.

- “We consider that this previous report supports our results, although the mechanisms underlying how IL-18BP causes a poor outcome in IPF remains unclear.” This is true but can the authors better explain what are the state of art about the knowledge of il-18 in the lung and the implications in ipf pathogenesis?

RESPONSE #18.

Thank you for your suggestion. We have speculated on two possible links between IL-18BP and poor outcome of IPF and these explanations were described in the next paragraph of the original manuscript. We have reformulated the paragraph structure for better understanding (Line 339 to 369).

COMMENT #19.

- “Despite these limitations, we believe that the results of our study are of significance in that we have demonstrated the utility of serum IL-18BP as a prognostic biomarker for IPF for the first time.” The final sentence resulted vague and inappropriate.

RESPONSE #19.

Thank you for your suggestion. As the reviewer pointed out, this sentence is vague and it overlapped with the descriptions in Conclusion section. Therefore, we deleted this sentence (Line 404).

Conclusion

COMMENT #20.

- The conclusions are inappropriate. Please better specify the role of our research avoiding vague sentences.

RESPONSE #20.

Thank you for your suggestion. We changed to a clearer and less ambiguous sentence (Line 408).

Reviewer #2: 

Introduction

COMMENT #1.

“Recently, it was reported that administration of IL- 18BP to a BLM-injury model improved lung fibrosis, but the trend of IL-18BP secreted in vivo is unclear.” Could the authors please provide a citation for the study that this statement applies to?

RESPONSE #1.

Thank you for your suggestion. We added the appropriate citation (Line 74).

Methods

Immunohistochemical staining:

COMMENT #2.

Could the authors please provide further information on the surgical lung biopsies? Were these tissues reviewed by a pathologist? What were the inclusion/exclusion criteria? How many tissues were collected? What dates were these tissues collected from? Are demographic/clinical characteristics available for these subjects?

RESPONSE #2.

We agree that the reviewer’s comments are important. First, all tissue samples were reviewed by pathologists to confirm the pathological diagnosis of UIP. Next, in principle, we tried to perform surgical lung biopsy for all patients without typical UIP pattern in HRCT. However, it was impossible frequently because of poor lung function or patients’ rejection. Actually, lung tissue was available in 9 patients. These explanations were added in “Immunohistochemical staining for IL-18BP” subsection (Line 80). Finally, we included biopsy date (days from the time of diagnosis) as well as demographic/clinical characteristics of these patients in S1 table.

Subjects for ELISA measurements:

COMMENT #3.

Could the authors please clarify what samples were collected from these subjects?

RESPONSE #3.

In accordance with the reviewers’ comment, we altered the description in this subsection to clarify what kind of samples were obtained (Line 101 and 107).

Results

Figure 1:

COMMENT #4.

How many tissues were stained? It would be beneficial to include images from other IPF and control subjects as well, to confirm findings across multiple samples.

COMMENT #5.

Please include a quantitative measure of staining to strengthen the findings.

RESPONSE #4 and 5.

As we mentioned in response to your Comment #2, lung tissue was available in nine of 86 IPF patients. Additionally, five samples were collected as control. We included two additional images of IPF patients in the revised Fig 1. As the reviewer pointed out in COMMENT#5, we performed quantitative measurement of IL-18BP expression using ImageJ Fiji software. These process were included in the Methods section and the results were included in the revised manuscript (Line 80 to 85, Line 92 to 96, Line 186, Line 189 to 192 and S1 Fig).

Figure 3:

COMMENT #6.

In panel c the authors state there is a positive correlation, but the plot demonstrates a negative correlation. Please adjust.

RESPONSE #6.

We agree that the description was mistaken. We revised it as the reviewer pointed out (Line 286).

Discussion

COMMENT #7.

The findings of this paper convey the significance of IL-18BP in both the circulatory and lung tissue compartments in IPF. Although the authors state it is expressed in bronchiole epithelial cells and alveolar macrophages in the IHC results, is there any existing literature on the cell types that IL-18BP is typically expressed in, in both the blood and lung tissue? It would be beneficial for the authors to include this, as well as how this may relate to the results of this study.

RESPONSE #7.

RESPONSE #7.

We agree that the reviewer’s comment is of critical importance. Unfortunately, we couldn’t find such literature, but in the Human Protein Atlas, an online bioinformatics database, marked expression of IL-18BP mRNA in granulocytes, monocytes and T-cells is reported. Therefore, we can speculate that the migrated inflammatory cells can be the main source of IL-18BP expression in the IPF lung. These points were included in Discussion section of the revised manuscript (Line 373 to 380).

Conclusion

COMMENT #8.

The authors state that IL-18BP “might be a potential therapeutic target of IPF.” It would be useful for the authors to further explain this in the discussion section, given what is known about its mechanism in conjunction with the findings of this manuscript.

RESPONSE #8.

We agree that the reviewer’s comment is reasonable, however, we have decided to delete this sentence in accordance with Comment #20 of Reviewer 1. We stated that IL-18BP “might be a potential therapeutic target of IPF” based on the works reported by Zhang and coworkers revealing that exogenous IL-18BP administration could attenuate bleomycin-induced lung fibrosis in murine models [14]. However, because this statement is not derived directly from our presenting study, we considered this statement as inappropriate as the conclusion of our manuscript (Line 408).

General

COMMENT #9.

The manuscript should be diligently edited for grammar and syntax errors.

RESPONSE #9.

In accordance with the reviewer’s comment, we would get English proofreading before resubmission.

Reviewer #3: This paper may be of interest to those researching IPF; a disease which requires better predictive biomarkers of disease prognosis.

COMMENT #1.

In the introduction the authors explain and highlight the need for early diagnosis of IPF and the requirement for predictive biomarkers of prognosis for clinical management. They explain how they arrived at their hypothesis (from a previous publication by this group) that IL-18BP may serve as a clinical biomarker for IPF. The methods are sufficiently detailed.

RESPONSE #1.

We acknowledge your kind feedback. 

COMMENT #2.

The authors suggest that IL-18BP is elevated in IPF lung tissue compared to non-cancerous lung tissue from lung cancer patients. They address the limitation of this analysis due to the number of subjects being relatively small, however n numbers are not provided. N numbers should be provided in the methods, results or figure legend and it should be made clear that the images shown in figure 1 are representative of a larger cohort.

RESPONSE #2.

We admit that the reviewer’s comment is reasonable. Actually, lung tissue was available in 9 patients with IPF. We included this information in Methods section and the background characteristics of these patients were presented in supplementary Table 1 in the revised manuscript (Line 80 to 83 and S1 Table). Further, we added some other representative pictures of lung tissue in Fig 1 (Line 186 and Fig 1).

COMMENT #3.

The authors show that both IL-18 and IL-18BP levels are increased in IPF patient serum compared to healthy controls and that serum IL-18BP may be more effective than IL-18 at discriminating IPF patients from healthy controls. The authors go on to show that serum IL-18 and IL-18BP levels correlate with each other in both IPF and healthy patients. The authors refer to ‘S1 fig’ to demonstrate this finding, however these graphs need titles to explain which refers to healthy and which refers to IPF data.

RESPONSE #3.

In accordance with the reviewer’s comment, we added titles at the top of the graphs (S2 Fig)

COMMENT #4.

The authors show that IPF serum IL-18BP and IPF BALF IL-18BP levels also correlate with each other, however due to the lack of healthy BALF the same comparison could not made for healthy patients and this should be considered as a limitation when making any conclusions from this. However, this does not particularly affect the overall findings of this manuscript.

RESPONSE #4.

We agree with the reviewer’s comment. From the view of medical ethics, we didn’t perform BAL in healthy subjects (Line 380).

COMMENT #5.

The authors describe in fig 3 that IL-18BP is inversely correlated with %DLco and %FVC which further supports IL-18BP as a biomarker of prognosis in IPF. In figure 3 they also use a Kaplan-Meier plot to illustrate that IPF patients with higher serum levels of IL-18BP (>5.72 ng/ml) showed poorer survival compared to those with lower IL-18BP levels. The authors refer to ‘S3 Fig’ to explain how they determined 5.72ng/ml as the optimal cut-off for predicting the 3 years survival of patients, however I do not understand how this figure explains this and further explanation is required.

RESPONSE 5.

In accordance with the reviewer’s comment, we altered the legend for S4 Fig as well as the description in “High serum IL-18BP was associated with poor prognosis and impaired pulmonary function” subsection of Result section (Line 261 and 580).

COMMENT #6.

Finally, the authors use a mouse model of lung fibrosis to corroborate their findings in humans, illustrating that increases in IL-18 and IL-18BP are associated with increased hydroxyproline expression in lung tissue. I question why they felt that they needed to use a mouse model to confirm what they had already demonstrated in humans. The advantage of using a mouse model here would be to try and understand a pathogenic role for IL-18BP in lung fibrosis however this was not done. For example, in the discussion they speculate two potential roles for IL-18BP in fibrosis; a pathogenic role via inhibition of IL-37 and a non-pathogenic role/protective role simply as a consequence of increased IL-18 expression. Both of these are valid hypotheses that could quite easily have been investigated in the mouse model used. Did the authors look for increased expression of IL-37 in this model to support this?

RESPONSE #6.

We admit that our mouse model experiments were insufficient to fully understand a pathogenic role for IL-18BP in lung fibrosis. Our primal hypothesis was that the increased expression of IL-18BP in IPF patients was not the cause of fibrosis, but the result of the increased expression of IL-18, a profibrotic factor. We considered that the unimodal elevation of BALF IL-18 followed by the bimodal elevation of BALF IL-18BP in our murine models might support this hypothesis in part. To explain these points more clearly, we reconstructed the 3rd and 4th paragraphs in Discussion section (Line 339 to 369). Of course, as the reviewer pointed out, we should have checked the expression of IL-37, but it was impossible mainly due to our financial limitation. We also included these weakness of this study in the limitation of the study (Line 403).

COMMENT #7.

The authors conclude that IL-18BP is elevated in IPF and may be a predictive biomarker of prognosis. They go on to suggest that IL-18BP may be a key molecule in the development and progression of IPF and might be a potential therapeutic target. I don’t see any evidence in this paper that supports the claim that IL-18BP has a pathogenic role in IPF or that by inhibiting it would impact the course of disease (again where the mouse model could have been used to explore this mechanism). We know from experience that simply because something is elevated in disease it does not mean it is pathogenic or make it a therapeutic target. The known actions of IL-18BP could equally support an attempt by the body to protect against disease progression in patients with rapidly progressing disease.

RESPONSE #7.

As the reviewer pointed out, we agree that the statement in Conclusion section was inappropriate and misleading. What we have demonstrated in this study was simply that IL-18BP was elevated in disease and it can be a prognostic biomarker for IPF. Therefore, we changed the statement in Conclusion section (Line 407).

Overall, the paper is sound with these minor revisions:

COMMENT #8.

1) Include n numbers for figure 1 and explain images shown are representative.

RESPONSE #8.

We investigated the lung tissue derived from nine patients with IPF. In accordance with the comment of reviewer 2, we included three representative images from these nine patients in the revised version (Line 80 to 83, Line 186 and Fig 1).

COMMENT #9.

2) Add graph titles to ‘S1 fig’ to discriminate between IPF and healthy controls.

RESPONSE #9.

In accordance with the reviewer’s comment, we added titles at the top of the graphs (S2 Fig)

COMMENT #10.

3) Include lack of healthy BALF as a limitation to this study when discussing correlations between IL-18BP in serum an BALF.

RESPONSE #10.

We agree with the reviewer’s comment. From the view of medical ethics, we didn’t perform BAL in healthy subjects (Line 380).

COMMENT #11.

4) Further explanation of how they determined 5.72ng/ml IL-18BP as the optimal cut-off for predicting the 3 year survival of patients, as it is not clear from ‘S3 fig’.

RESPONSE #11.

In accordance with the reviewer’s comment, we altered the legend for S3 Fig as well as the description in “High serum IL-18BP was associated with poor prognosis and impaired pulmonary function” subsection of Result section (Line 261 and 580).

COMMENT #12.

5) It is a shame that the mouse model wasn’t used to investigate the role of IL-18BP in fibrosis. Did the authors consider looking at IL-37 in this model to support their hypothesis in the discussion?

RESPONSE #12.

We admit that our mouse model experiments were insufficient to fully understand a pathogenic role for IL-18BP in lung fibrosis. Our primal hypothesis was that the increased expression of IL-18BP in IPF patients was not the cause of fibrosis, but the result of the increased expression of IL-18, a profibrotic factor. We considered that the unimodal elevation of BALF IL-18 followed by the bimodal elevation of BALF IL-18BP in our murine models might support this hypothesis in part. To explain these points more clearly, we reconstructed the 3rd and 4th paragraphs in Discussion section (Line 339 to 369). Of course, as the reviewer pointed out, we should have checked the expression of IL-37, but it was impossible mainly due to our financial limitation (Line 403).

COMMENT #13.

6) If the authors have any evidence to support a pathogenic role of IL-18BP in IPF it should be included to support the claim that “IL-18BP may be a key molecule in the development and progression of IPF and might be a potential therapeutic target of IPF” rather than it is simply elevated in disease and a potential clinical biomarker which is otherwise what they have demonstrated. Alternatively, make it clear that this is a hypothesis of this group and further experiments are required. Literature suggests that IL-18BP is potentially protective in fibrosis and its inhibition could actually exacerbate disease.

RESPONSE #13.

As the reviewer pointed out, we agree that the statement in Conclusion section was inappropriate and misleading. What we have demonstrated in this study was simply that IL-18BP was elevated in disease and it can be a prognostic biomarker for IPF. Therefore, we changed the statement in Conclusion section (Line 407).

---

## [Decision Letter · Decision Letter 1]

19 May 2021

IL-18 binding protein can be a prognostic biomarker for idiopathic pulmonary fibrosis

PONE-D-20-35981R1

Dear Dr. Horimasu,

We’re pleased to inform you that your manuscript has been judged scientifically suitable for publication and will be formally accepted for publication once it meets all outstanding technical requirements.

Kind regards,

Gernot Zissel, Ph.D.

Academic Editor

PLOS ONE

Additional Editor Comments (optional):

Reviewers' comments:

Reviewer's Responses to Questions

**Comments to the Author**

1. If the authors have adequately addressed your comments raised in a previous round of review and you feel that this manuscript is now acceptable for publication, you may indicate that here to bypass the “Comments to the Author” section, enter your conflict of interest statement in the “Confidential to Editor” section, and submit your "Accept" recommendation.

Reviewer #2: All comments have been addressed

Reviewer #3: All comments have been addressed

2. Is the manuscript technically sound, and do the data support the conclusions?

Reviewer #2: Yes

Reviewer #3: Yes

3. Has the statistical analysis been performed appropriately and rigorously? 

Reviewer #2: Yes

Reviewer #3: I Don't Know

4. Have the authors made all data underlying the findings in their manuscript fully available?

Reviewer #2: Yes

Reviewer #3: Yes

5. Is the manuscript presented in an intelligible fashion and written in standard English?

Reviewer #2: Yes

Reviewer #3: Yes

6. Review Comments to the Author

Reviewer #2: The authors have addressed all the previous concerns and this reviewer finds the manuscript much improved.

Reviewer #3: An interesting piece of work that presents evidence to support a role of IL-18BP as a prognostic marker in IPF. Further investigations into understanding the potential pathogenic mechanism of the IL-18 pathway in IPF will be of interest. All comments have been responded to sufficiently.

7. PLOS authors have the option to publish the peer review history of their article (what does this mean?). If published, this will include your full peer review and any attached files.

Reviewer #2: No

Reviewer #3: No

---

## [Editor Report · Acceptance letter]

27 May 2021

PONE-D-20-35981R1 

IL-18 binding protein can be a prognostic biomarker for idiopathic pulmonary fibrosis 

Dear Dr. Horimasu:

I'm pleased to inform you that your manuscript has been deemed suitable for publication in PLOS ONE. Congratulations! Your manuscript is now with our production department. 

Kind regards, 

on behalf of

Prof. Dr. Gernot Zissel 

Academic Editor

PLOS ONE